# Boosting Graph Convolution with Disparity-induced Structural Refinement

Submission Id: 11

## ABSTRACT

Graph Neural Networks (GNNs) have expressed remarkable capability in processing graph-structured data. Recent studies have found that most GNNs rely on the homophily assumption of graphs, leading to unsatisfactory performance on heterophilous graphs. While certain methods have been developed to address heterophilous links, they lack more precise estimation of high-order relationships between nodes. This could result in the aggregation of excessive interference information during message propagation, thus degrading the representation ability of learned features. In this work, we propose a Disparity-induced Structural Refinement (DSR) method that enables adaptive and selective message propagation in GNN, to enhance representation learning in heterophilous graphs. We theoretically analyze the necessity of structural refinement during message passing grounded in the derivation of error bound for node classification. To this end, we design a disparity score that combines both features and structural information at the node level, reflecting the connectivity degree of hopping neighbor nodes. Based on the disparity score, we can adjust the aggregation of neighbor nodes, thereby mitigating the impact of irrelevant information during message passing. Experimental results demonstrate that our method achieves competitive performance, mostly outperforming advanced methods on both homophilous and heterophilous datasets.

## CCS CONCEPTS

• **Mathematics of computing → Graph algorithms**; • **Computing methodologies → Neural networks**.

## KEYWORDS

Graph neural network, homophily and heterophily, structural learning, message passing.

## 1 INTRODUCTION

Graph-structured data are prevalent in the real world, exemplified by social networks and molecular structures. To effectively address such non-Euclidean data, Graph Neural Networks (GNNs) have emerged as powerful tools, extensively applied across various domains, including traffic prediction [8, 12, 41], molecular exploration [13, 35, 37], classification and clustering [10, 23, 32]

*Conference '25, April 28, 2025, Sydney, Australia*
© 2025 Association for Computing Machinery.
ACM ISBN 978-x-xxxx-xxxx-x/YY/MM...$15.00
https://doi.org/XXXXXXX.XXXXXXX

and others [1, 20, 24]. As a pivotal stage of GNNs, message passing transforms and disseminates information through the graph's topology, significantly enhancing the expressiveness of learned feature. Most GNNs [6, 11, 16] are designed under the homophily assumption, where nodes with similar labels or features tend to be connected. However, real-world applications frequently involve highly heterophilous graphs, such as in the case of amino acids of different types forming connections. Consequently, many previous GNNs proposed for homophilous networks, such as GEGCN [21], JKNet [38] and APPNP [15], struggle to effectively capture heterophily, resulting in an unsatisfactory performance on heterophilous networks.

Real-world graphs typically contain both homophilous and heterophilous edges. A graph is considered homophilous when the former outnumber the latter; otherwise, it is viewed as heterophilous. Recent studies have revealed that the smoothing operation inherent in GNNs can generate similar node features for nodes with different labels, when applied to graphs with heterophily [7, 19, 26]. To mitigate the negative impact of this issue on node classification tasks, various designs have been developed to enhance the discriminative capabilities of GNNs in heterophilous scenarios. One typical strategy is to construct augmented graphs by introducing additional semantics to prevent nodes from different classes from adopting similar representations. For instance, Huang et al. [9] utilized known edge labels to identify other links, thus facilitating message passing by removing all heterophilous edges. Pei et al. [29] redefined graph convolution by utilizing geometric relationships in the latent space. These methods primarily focus on the detrimental effects of heterophilous connections. However, they often overlook the potential advantages of effectively identifying and leveraging heterophilous edges.

Another established approach involves learning signed edges to cluster similar nodes while repelling dissimilar ones, which relocates edges and facilitates message passing adopting the whole graph topology. In this context, homophilous relationships are assigned positive signs, whereas heterophilous connections are designated negative signs. For instance, graph attention functions were used to compute signed edges such that node representations were better learned [2, 40]. To better define signed edges, [4, 39] designed both low-pass and high-pass filters to differentiate between various connections. In graphs with high heterophily, direct neighbors often exhibit greater heterophily than multi-hop neighbors. Consequently, these algorithms aggregate high-order information by applying signed convolutional filters multiple times, effectively utilizing edges with assigned signs and weights for information propagation and fusion. Nevertheless, these methods encounter several limitations in the context of heterophilous graphs: i) They solely rely on node features to infer node relationships, neglecting structure information, which can easily establish inaccurate estimation; ii) The interplay between the discriminative capacity

of models and the nature of homophilous/heterophilous graphs remains unclear.

To address the issues aforementioned, we propose a Disparity-induced Structural Refinement (DSR) framework with the integration integrated with GNN, named DSR-GNN, aimed at enhancing node representations in heterophilous graphs. Grounded in the theory of error bound for node classification, we first conduct a theoretical analysis of the factors affecting the model's capacity to handle heterophilous graphs, underscoring the necessity of exploring refined graph structures. Our proposed architecture integrates two collaborative steps: assessing node relationships and performing message passing on refined graphs. In the initial step, we evaluate high-order node relationships in graphs by calculating a disparity score that combines distances of aggregated features and differences in homophily ratios. Subsequently, the score drives the construction of layer-wise adjacency edges by removing links with significant disparity. This refinement process ensures that message passing is conducted on graphs with minimized interference from irrelevant high-order information. Notably, the updated node representations from message passing can, in turn, update the disparity score. Together, these two collaborative steps facilitate the attainment of more discriminative node representations.

Our contributions can be summarized in three aspects:

i) We propose a disparity-induced structural refinement framework, theoretically dissecting the relationship between model capacity and homo/heterophilous ratios, to enhance representation learning in heterophilous graphs.

ii) We propose a disparity score that integrates both features and structural information at the node level, facilitating structural refinement and mitigating the impact of irrelevant information during message passing.

iii) Extensive experiments demonstrate that the proposed model achieves state-of-the-art performance on heterophilous graphs and competitive accuracy on homophilous networks.

**Overview.** In the remainder of this paper, we first introduce the primary preliminaries used in the paper in Section 2. Following this, Section 3 analyzes the theoretical background of our research, and Section 4 presents our framework DSR-GNN. Finally, we conduct extensive experiments in Section 5 and conclude our work in Section 6.

## 2 PRELIMINARIES

### 2.1 Notations

Given an undirected graph $\mathcal{G}(V, E)$ with $N$ nodes ($\{v_i \in V|_{i=1}^N\}$) and $e$ edges, where $V = V_{\text{lab}} \cup V_{\text{unlab}}$ with labeled node set $V_{\text{lab}}$ and unlabeled node set $V_{\text{unlab}}$, and $e_{ij} \in E$ denotes the edge between the $i$-th and $j$-th nodes. The topological relationships among nodes are expressed as $\mathbf{A} \in \mathbb{R}^{N \times N}$, where $A_{ij} = 1$ if nodes $i$ and $j$ are connected, 0 otherwise. Moreover, $\hat{\mathbf{A}} = \mathbf{A} + \mathbf{I}$ stands for $\mathbf{A}$ with added self-loops, while $\widetilde{\mathbf{A}} = \hat{\mathbf{D}}^{-1/2}\hat{\mathbf{A}}\hat{\mathbf{D}}^{-1/2}$ denotes the symmetric normalized adjacency matrix. Note that the renormalization trick on the adjacency matrix is used to prevent gradient explosion. Here, $\hat{\mathbf{D}}$ is the diagonal degree matrix, where $\hat{D}_{ii} = \sum_{j=1}^N \hat{A}_{ij}$. $\mathbf{X} \in \mathbb{R}^{N \times d}$ indicates node features, in which $\mathbf{x}_i$ with $d$-dimensions is the feature vector of the $i$-th node. Among the $N$ nodes, $N_{\text{lab}}$ nodes are labeled,

with their labels captured in the ground truth matrix $\mathbf{Y} \in \mathbb{R}^{N_{\text{lab}} \times C}$, where $C$ is the number of classes, and each row $\mathbf{y}_i$ of $\mathbf{Y}$ is a one-hot vector representing the label of node $v_i$.

### 2.2 Node-level Homophily and Heterophily

Given a set of nodes with labels, the homophily ratio of each node calculates the tendency of the node to have the same label as its neighbors. Considering node $v_i$, we assume its neighbor set as $\mathcal{N}_i$, then the homophily ratio of node $v_i$ is defined as: $h_i^+ = \frac{|\{\mathbf{y}_i = \mathbf{y}_j | v_j \in \mathcal{N}_i\}|}{|\mathcal{N}_i|}$. $h_i^+$ ranges in $[0, 1]$, with values close to 1 indicating high homophily (or low heterophily) and values close to 0 indicating the opposite. Corresponding, the heterophily ratio $h_i^- = 1 - h_i^+$. Therefore, the node-level homophily in the graph $\mathcal{G}$ can be measured by $\mathcal{H}(\mathcal{G}) = \frac{\sum_{i=1}^N h_i^+}{N}$. Many previous works have explored heterophily using the above node-level homophily metric and have proposed various approaches, such as signed edges, to reduce the impact of confusing information brought by non-similar neighbors [4, 39].

### 2.3 Graph Neural Network for Semi-supervised Classification

The core of GNN is the message passing, which collects the neighborhood information to update node representations. Consider a GNN with $L$ layers, where the output of the $l$-th layer is given by: $\mathbf{h}_i^{(l)} = \sigma\Big(\text{Aggregate}(\{\mathbf{h}_j^{(l-1)}|\hat{A}_{ij} = 1\})\mathbf{\Theta}^{(l)}\Big)$. Here, $\mathbf{\Theta}^{(l)}$ is the trainable parameter matrix of the $l$-th layer and $\sigma(\cdot)$ indicates the ReLU($\cdot$) or Softmax($\cdot$) activation function. After gaining the final representation $\mathbf{H}^{(L)}$, the cross-entropy loss consisted of $\mathbf{H}^{(L)}$ and $\mathbf{Y}$ is attained: $\mathcal{L}_{ce} = -\sum_{i \in \Omega} \sum_{j=1}^C Y_{ij} \ln(H_{ij}^{(L)})$. Here, $\Omega$ is the set of labeled samples. To simplify the model parameter, some models [3, 15] firstly use the fully connected neural network on the feature matrix $\mathbf{X}$ to generate the hidden state features $\mathbf{H}^{(0)}$ and then propagate them via the message passing. Their updating rule can be defined as: $\mathbf{H}^{(l)} = \sigma(\widetilde{\mathbf{A}}\mathbf{H}^{(l-1)} + \mathbf{H}^{(0)}), \mathbf{H}^{(0)} = \Phi_\theta(\mathbf{X})$, where $\theta$ is the parameter set of the neural network $\Phi$.

## 3 THEORETICAL DISPARITY ANALYSIS

Classic graph convolution methods typically assume that nodes belonging to the same class are more likely to be connected, which fails to hold in heterophilous graphs. To investigate the factors affecting the model's ability to differentiate between nodes, we derive an error bound for node classification, which can boost the design of effective message-passing for heterophilous graphs. Theoretically, drawing inspiration from PAC-Bayes analysis [5, 27], we delineate the key assumptions and definitions related to graph data and classifiers, followed by a thorough derivation of the error bound applicable to any unlabeled nodes.

DEFINITION 1. *Let's define a $L$-layer GNN classifier $f$, for node $v_i$, the prediction score is $f_i(\mathbf{X}, \mathcal{G}) = f(g_i(\mathbf{X}, \mathcal{G}); \mathbf{\Theta}^{(1)}, \mathbf{\Theta}^{(2)}, \cdots, \mathbf{\Theta}^{(L)})$, where $g$ denotes a feature aggregation function and $f$ is a ReLU-activated $L$-layer MLP with learnable parameters $\{\mathbf{\Theta}^{(l)}\}_{l=1}^L$. We assume that the maximum number of hidden units across all layers is $b$.*

DEFINITION 2. *For any node $v_i$, the distance of aggregated features from it to other node $v_j$ is defined as*

$$\epsilon_{ij} = \|g_i(\mathbf{X}, \mathcal{G}) - g_j(\mathbf{X}, \mathcal{G})\|_2. \tag{1}$$

DEFINITION 3. *Given a labeled node $v_j \in V_{lab}$ with label $y_j$, there exists a margin $\gamma \geq 0$ satisfing*

$$f_j(\mathbf{X}, \mathcal{G})[y_j] \leq \gamma + max_{c \neq y_j} f_j(\mathbf{X}, \mathcal{G})[c], \tag{2}$$

*where $f_j(\mathbf{X}, \mathcal{G})[\cdot]$ is to take an element of the predicted probability vector (w.r.t classifier).*

DEFINITION 4. *The expected loss $\mathcal{L}_i^\gamma(f)$ of the classifier $f$ on $v_i$ for a margin $\gamma$ and any distribution $\mathcal{D}$ is defined as [25, 28]:*

$$\mathcal{L}_i^\gamma(f) := \mathbb{P}_{v_i \sim \mathcal{D}} \Big[ f_i(\mathbf{X}, \mathcal{G})[y_i] \leq \gamma + max_{c \neq y_i} f_i(\mathbf{X}, \mathcal{G})[c] \Big]. \tag{3}$$

*The empirical loss is denoted as $\hat{\mathcal{L}}_i^\gamma(f)$ that is the empirical estimate of the expected loss.*

According to the above definitions, the error bound for semi-supervised node classification is illustrated as below. It aims to bound the expected loss $\mathcal{L}_i^0$ of classifier on the unlabeled node $v_i$ for a margin 0. Here, the empirical loss on the labeled node $v_j$ for a margin $\gamma$ is denoted as $\hat{\mathcal{L}}_j^\gamma$.

THEOREM 1 (ERROR BOUND FOR UNLABELED NODE CLASSIFICATION). *Let $f$ be a classifier in the classifier family $\mathcal{F}$ with learnable parameters $\{\Theta^{(l)}\}_{l=1}^L$ that conform with the normal distribution, then for any unlabeled node $v_i$ and $\gamma \geq 0$, we have*

$$\mathcal{L}_i^0(f) \leq \hat{\mathcal{L}}_j^\gamma(f) + O\Big( \frac{C\rho}{\sqrt{2\pi}\sigma}(\epsilon_{ij} + \rho|h_i^+ - h_j^+|)$$
$$+ \frac{\sum_{l=1}^L \|\Theta^{(l)}\|_F^2}{\sigma^2} \Big), \tag{4}$$

*where $\sigma = \min\Big( \frac{(\gamma/8\epsilon_{ij})^{1/L}}{\sqrt{2b(1+\ln(2bL))}}, \frac{\gamma}{84LB_i\beta^{L-1}\sqrt{b\ln(4bL)}} \Big)$, $B_i = \|g_i(\mathbf{X}, \mathcal{G})\|_2$, $h_i^+$ denotes the homophily ratio of node $v_i$ and $\rho$ is original feature separability of nodes.*

PROOF. The proof is deferred to **Appendix**. □

This theorem elucidates that the primary factors influencing the error bound are the distance of aggregated feature $\epsilon_{ij} = \|g_i(\mathbf{X}, \mathcal{G}) - g_j(\mathbf{X}, \mathcal{G})\|_2$ and the disparity in homophily ratios[1]: $|h_i^+ - h_j^+|$. Conventional GNN methods primarily emphasize minimizing the distance of aggregated feature to enhance representation learning, often neglecting the significance of homophily ratios, which fundamentally reflect the underlying graph structure.

REMARK 1. *In previous studies, two key aspects of debate have emerged regarding heterophily (conversely homophily) in graph convolution. One perspective asserts that heterophily is detrimental to message passing, as connections between nodes of different classes can lead to mixed features, resulting in indistinguishable node representations [4, 42]. Another viewpoint posits that heterophilous edges can be advantageous, as they not only enhance the differentiation of inter-class information but also facilitate long-distance message passing [9, 39]. Different from them, according to Theorem 1, we should consider the **disparity** of structure ($|h_i^+ - h_j^+|$) and feature*

---

$(\epsilon_{ij})$ *between nodes during message passing to balance advantages and disadvantages of heterophilous links, rather than simply adjusting heterophily/homophily.*

To this end, we attempt to devise an effective structural adjustment strategy that leverages the disparity of homophily ratios as well as the distance of aggregated features. This strategy aims to reduce error bounds and enhance the discriminative capacity of the model. The central idea is to refine graph structures to mitigate the influence of irrelevant high-order information while facilitating more meaningful message passing, thereby improving the model's discernibility.

## 4 DISPARITY-INDUCED STRUCTURAL REFINEMENT

In this section, we present the disparity-induced structural refinement method, designed for integration with graph neural network, inspired by the insights from Theorem 1. This method consists of three critical steps: evaluating edge signs, computing the disparity score, and adjusting message propagation. We finally aggregate the node representations updated across all refined graphs to obtain the final predicted results.

### 4.1 Assign Homo/Heterophile Edges

In order to estimate the node-level homophily ratio, it is essential to annotate the homophily and heterophily properties of the $k$-hop neighboring nodes surrounding a given node. It sometimes aligns with the concept of signed edges, which could enhance the purity of neighbor information gathered during message aggregation. Specifically, we assign positive signs to edges connecting nodes of the same class (i.e., homophilous edges) and negative signs to those linking nodes of distinct categories (i.e., heterophilous edges). By doing so, the use of signed edges allows the model better capture graph structure, thereby improving discrimination between nodes belonging to distinct classes. Incorrectly assigning a negative sign to a homophilous edge or a positive sign to a heterophilous edge can not only hinder model performance but may also lead to degradation, as demonstrated in GGCN [39]. Therefore, accurately matching signs to edges is paramount. To address this, we propose a pre-training process that enhances the accuracy of signed edges, effectively mitigating the influence of noise in the raw data, rather than merely relying on the cosine similarity of the original node features as utilized in [39].

Concretely, we learn a way of generating signed edges from the training set, leveraging the set of labeled nodes. Let $E_{lab}$ denotes the set of edges just that exist solely between labeled samples. We can then define a signed matrix $\mathbf{W} \in \mathbb{R}^{N \times N}$ restricted to the labeled edges during the training phase, with elements drawn from the set $\{-1, 0, 1\}$. Formally,

$$W_{ij} = \begin{cases} 1, & \text{if } v_i, v_j \in V_{lab} \ \& \ e_{ij} \in E_{lab} \ \& \ \mathbf{y}_i = \mathbf{y}_j, \\ -1, & \text{if } v_i, v_j \in V_{lab} \ \& \ e_{ij} \in E_{lab} \ \& \ \mathbf{y}_i \neq \mathbf{y}_j, \\ 0, & \text{otherwise.} \end{cases} \tag{5}$$

Eq. (5) indicates the true signed edges for the training phase. In order to learn a prediction model of signed edges, we concatenate the representations of two connected nodes to form the feature of the corresponding edge. Formally, the feature of the edge connecting

---

[1] $|h_i^+ - h_j^+| = |h_i^- - h_j^-|$, as $h_i^+ + h_i^- = 1$.

nodes $v_i$ and $v_j$ is denoted as $[\mathbf{x}_i||\mathbf{x}_j]$, where $||$ represents vector concatenation. To predict the sign of each edge, we input these edge features into a multi-layer perceptron (MLP) as follows,

$$\widetilde{W}_{ij} \leftarrow \text{sgn}(\text{Tanh}(\text{MLP}([\mathbf{x}_i||\mathbf{x}_j]))), \quad (6)$$

where "sgn" is the sign function, "Tanh" is the hyperbolic tangent activation function that maps values to the range [-1, 1]. We optimize the MLP through gradient backpropagation on the Mean Squared Error (MSE) loss, defined as: $\mathcal{L}_{mse} = \frac{1}{|E_{\text{lab}}|} \sum_{(v_i, v_j) \in E_{\text{lab}}} (W_{ij} - \widetilde{W}_{ij})^2$. Based on the learnt prediction model, we can generate a signed matrix $\widetilde{\mathbf{W}}$, whose elements are whether predicted signs of unsigned edges (in testing) or true signed edges (in training).

Remark 2. *The pre-training procedure of edge assignment described above utilizes existing training edges, whereby labeled samples are interconnected. In the absence of such conditions, our model can proceed without pre-training, instead estimating edge signs based on the similarity between nodes. Following the prediction of edge signs using $\widetilde{\mathbf{W}}$, we can gain the final representations through a step-wise integration of various high-order neighbor signals.*

## 4.2 Compute Disparity Scores

Building on Theorem 1, we conclude that the classification error is primarily influenced by the distance between aggregated features and the disparity in homophily ratios. To address this, we incorporate these two critical factors into a unified disparity score, the calculation of which is detailed in this subsection.

After learning the signed matrix $\widetilde{\mathbf{W}}$ with Eq. (6), the $k$-hop homophily ratio of node $v_i$ is defined as

$$h_i^{(k)} = \frac{|\{v_j | v_j \in \mathcal{N}_i^{(k)}, \widetilde{W}_{ij}^{(k)} > 0\}|}{|\mathcal{N}_i^{(k)}|}, \quad (7)$$

where the superscript $(k)$ denotes the $k$-hop neighbors[2]. Hereby, we can compute the $l$-hop disparity score between node $v_i$ and its neighbor $v_j$ as follows,

$$S_{ij}^{(l)} = \|\mathbf{h}_i^{(l-1)} - \mathbf{h}_j^{(l-1)}\|_2 + |h_i^{(l)} - h_j^{(l)}|. \quad (8)$$

The first term represents the aggregated-feature distance with $\mathbf{h}_i^{(l)} = \text{ReLU}\left(g(\{\mathbf{h}_j^{(l-1)} | v_j \in \mathcal{N}_i^{(l)}\})\right)$, where $g(\cdot)$ is an aggregation function. The second term encapsulates the disparity in homophily ratios, reflecting the differences of neighbor substructure surrounding nodes $v_i$ and $v_j$. The score reflects the disparity between a given node and its the $l$-hop neighbor nodes, both in terms of feature and structure spaces, thereby serving to guide message propagation along accurate paths for obtaining discriminative node representations.

## 4.3 Adjust Message Propagation

According to disparity scores from Eq. (8), we adjust the aggregated neighboring nodes to mitigate the latent noise from surrounding neighbor information. Formally, for a given node $v_i$, the aggregated

---

[2]Different hopping levels utilize distinct sign matrices.

nodes in the $l$-th layer are defined as,

$$\mathcal{A}_i^{(l)} := \{v_j | v_j \in \mathcal{N}_i^{(l)} \wedge S_{ij}^{(l)} \leq \tau_i^{(l)}\}, \quad (9)$$

$$\text{s.t.,} \quad \tau_i^{(l)} = \frac{1}{|\mathcal{N}_i^{(l)}|} \sum_{v_j \in \mathcal{N}_i^{(l)}} S_{ij}^{(l)},$$

where $\tau_i^{(l)}$ represents the average score between node $v_i$ and its $l$-order neighbors. The construction of $\{\mathcal{A}_i^{(l)}\}_{l=1}^L$ is guided by disparity scores, preserving high-order neighbors that exhibit minimal differences to avoid the influence of irrelevant high-order information. Thus, given a set $\mathcal{A}_i^{(l)}$ containing neighbors of node $v_i$ to be aggregated in the $l$-th layer, we can perform message passing during graph convolution.

The message aggregation for the $l$-th layer is defined as follows,

$$\mathbf{h}_i^{(l)} =$$
$$\text{gCov}\left( \sum_{v_j \in \{v_i\} \cup \mathcal{A}_i^{(l)}} \widetilde{W}_{ij}^{(l)} (D_{ii}^{(l)} D_{jj}^{(l)})^{-1/2} \mathbf{h}_j^{(l-1)}, \Theta_1 \right), \quad (10)$$

where $l = 1, \cdots, L$, $\mathbf{h}_i^{(0)}$ is gained using a fully-connected neural network with parameter $\Theta_1 \in \mathbb{R}^{d \times m}$ on $\mathbf{x}_i$, and "gCov" denotes a conventional graph convolution layer followed by a ReLU activation function. Here, $D_{ii}^{(l)}$ represents the degree of node $i$, calculated from the aggregation neighbor structure $\mathcal{A}^{(l)}$ for the normalization purpose. Note that message aggregation is applied solely to the refined graph structures $\mathcal{A}^{(l)}$ besides the node itself. Furthermore, the representation $\mathbf{h}_i^{(l)}$, obtained by aggregating messages from $\mathcal{A}_i^{(l)}$, can iteratively update the disparity scores used in the $(l+1)$-th layer.

Through multi-layer message propagation, the final output can be derived by aggregating features from all layers as:

$$\hat{\mathbf{y}}_i = \text{Softmax}\left( \left( \sum_{l=0}^{L} \lambda_l \mathbf{h}_i^{(l)} \right) \Theta_2 \right), \quad (11)$$

where $\lambda_l$ is a learnable parameter that indicates the importance of features from each layer, and $\Theta_2 \in \mathbb{R}^{m \times c}$ is a weight matrix optimized for predicting class scores. The final outputs are then combined with the labels from the training data to compute the cross-entropy loss, facilitating model optimization.

In conclusion, we begin by learning a signed matrix that reflects the homophily and heterophily of links, utilizing edges between labeled samples. Subsequently, based on the node-level homophily ratios provided by the signed matrix and the aggregated features obtained by graph convolution, we compute the disparity score revealing high-order relationships between nodes. The disparity score explores key links from both node features and structures to obtain improved graph structures. Message propagation and aggregation are then performed on these refined graphs to derive discriminative node representations. Network parameters are optimized through backpropagation of the cross-entropy loss consisted of the final output and the labels from the training data. Algorithm 1 summarizes the updating process of variables. The network is implemented in Pytorch and uses GPU acceleration to boost training efficiency.

---

**Algorithm 1:** DSR-GNN

**Input:** Node features $\{\mathbf{x}_i \in \mathbb{R}^d\}_{i=1}^N$, original adjacency matrix $\mathbf{A}$, the signed matrix $\widetilde{\mathbf{W}}$, ground truth matrix $\mathbf{Y}$, layer number $L$.

**Output:** The predicted class label.

1   Initialize parameters $\boldsymbol{\Theta}_1, \boldsymbol{\Theta}_2, \{\lambda_l\}_{l=0}^L$;

2   $\mathbf{h}_i^{(0)} = \text{ReLU}(\mathbf{x}_i \boldsymbol{\Theta}_1)$;

3   **for** $l = 1 \rightarrow L$ **do**

4      Calculate $S_{ij}^{(l)}$ with Eqs. (7) and (8);

5      Compute $\tau_i^{(l)} = \frac{1}{|\mathcal{N}_i^{(l)}|} \sum_{v_j \in \mathcal{N}_i^{(l)}} S_{ij}^{(l)}$ and obtain $\mathcal{A}_i^{(l)}$ with Eq. (9);

6      Update $\mathbf{h}_i^{(l)}$ with Eq. (10);

7   Obtain $\hat{\mathbf{y}}_i = \text{Softmax}\left(\left(\sum_{l=0}^L \lambda_l \mathbf{h}_i^{(l)}\right) \boldsymbol{\Theta}_2\right)$;

8   Update parameters via backpropagation of the cross-entropy loss consisting of $\mathbf{Y}$ and $\hat{\mathbf{Y}}$ from the training data;

9   **return** The predicted class label of the $i$-th node is given by $\arg\max \hat{\mathbf{y}}_i$.

---

## 4.4 Connect to Other Methods

**DSR-GNN Vs. DropEdge.** DropEdge [30] constructs an augmented adjacency matrix by randomly removing partial edges, and the matrix is shared by all layers. Its strategy is sample and intuitive, but it doesn't fully leverage the inherent data structure of the graph. Compared to DropEdge, DSR-GNN uses the disparity score to guide the sampling procedure, ensuring that layer-wise adjacency matrices more accurately capture high-order relationships between nodes.

**DSR-GNN Vs. GPR-GNN.** Both GPR-GNN [4] and DSR-GNN adopt weighted summation to generate the final node representations. Differently, the key component of GPR-GNN lies in exploring learnable weights to adapt to the homophily or heterophily patterns within the graph; while DSR-GNN focuses on leveraging signed edges and structural refinement techniques. These schemes balance the topological propagation abilities with the inherent heterophily in the graph, resulting in more discriminative node representations.

**DSR-GNN Vs. GGCN.** GGCN [39] proposes two edge correction strategies based on the theoretical analysis, including structure-based and feature-based methods. The former rescales edge weights to satisfy the required node degree conditions; while DSR-GNN emphasizes exploring node-level high-order homophily structures. Moreover, its feature-based method leverages cosine similarity to gain edge signs, but DSR-GNN adopts a pre-training scheme to predict signs for unknown edges.

## 5 EXPERIMENTS

In this section, we construct a series of experiments to assess the effectiveness of DSR-GNN. Our model is implemented in PyTorch on a workstation with AMD Ryzen 9 5900X CPU (3.70GHz), 64GB RAM and RTX 3090GPU (24GB caches). We answer several key questions via experiments:

- **Q1:** How does DSR-GNN perform on both homophilous and heterophilous datasets?
- **Q2:** What is the impact of signed edges on model performance in heterophilous and homophilous graphs?
- **Q3:** How can we empirically verify the influence of structural refinement driven by the disparity score on performance?
- **Q4:** In what ways does the refined graph differ structurally from the original graph?
- **Q5:** How does high-order information affect the final representations learned by DSR-GNN?

## 5.1 Experimental Setups

Datasets. To validate the performance of DSR-GNN, we use three homophilous datasets: Cora, Citeseer and Pubmed [31], which are citation networks where nodes represent publications and edges correspond to citation links. Additionally, we test on four heterophilous datasets: Texas, Wisconsin, Cornell, and Actor [29]. In the Texas, Wisconsin, and Cornell datasets, nodes represent webpages, and edges denote hyperlinks between them. For the Actor dataset, each node represents an actor, with edges indicating co-occurrence on the same Wikipedia page. A summary of the dataset statistics is provided in Table 1.

Competitors. We compare DSR-GNN against 13 algorithms: 1) A baseline: 2-layer MLP; 2) Two classic GNN models: vanilla GCN [14] and GAT [34]; 3) Two recent models performing well on homophilous graphs: GCNII [3] and GCNet [36]; 4) Seven advanced models designed specifically to handle heterophily: FAGCN [2], $\text{H}^2\text{GCN}$ [42], GPR-GNN [4], GGCN [39], ACM-GCN [22], LRGNN[18] and PCNet [17].

Experimental Settings. We exploit accuracy (ACC) as the evaluation metric to measure the model's performance in correctly classifying samples. For all datasets, we randomly split training/validation/testing samples into 48%/32%/20% of all samples. For heterophilous datasets, the learning rate, weight decay, dropout rate and number of hidden units are set to 0.01, 5e-4, 0.1 and 128, respectively. For homophilous graphs, the configurations are largely analogous, with the exception that the weight decay is set to 0. Each experiment is performed 10 times, and the mean and standard deviation are recorded. Our code is available at **https://anonymous.4open.science/r/DSR-GNN-5876**.

## 5.2 (Q1) Classification Results on Benchmark Datasets

Table 2 presents a comparison of test accuracy across all algorithms on real-world datasets with different homophily levels. From this table, we draw the following observations:

- DSR-GNN obtains the optimal and suboptimal performance on most datasets, particularly on heterophilous networks. On homophilous datasets, DSR-GNN maintains its competitive performance, which is within 1% of the best model.
- MLP is a solid baseline for handling heterophilous graphs, outperforming models with implicit homophily assumptions, such as GCNII and GCNet. This observation underscores that message passing over indistinguishable edges can negatively impact performance.

**Table 1: Benchmark dataset statics.**

| Datasets | #Nodes | #Edges | #Features | #Classes | #Training/Testing/Validation |
|----------|--------|--------|-----------|----------|------------------------------|
| Citeseer | 3,327 | 4,676 | 3,703 | 7 | 1,597/1,065/665 |
| Cora | 2,708 | 5,278 | 1,433 | 6 | 1,300/867/541 |
| Pubmed | 19,717 | 44,327 | 500 | 3 | 9,464/6,309/3,944 |
| Texas | 183 | 295 | 1,703 | 5 | 88/59/36 |
| Wisconsin | 251 | 466 | 1,703 | 5 | 121/80/50 |
| Cornell | 183 | 280 | 1,703 | 5 | 121/80/50 |
| Actor | 7,600 | 26,752 | 931 | 5 | 3,648/2,432/1,520 |

**Table 2: Node classification results on real-world datasets: Mean accuracy (%) ± Standard deviation (%). Best performance is highlighted in bold, and runner-up accuracy is highlighted in underline.**

| Methods/Datasets | Citeseer | Cora | Pubmed | Texas | Wisconsin | Cornell | Actor |
|------------------|----------|------|--------|-------|-----------|---------|-------|
| MLP | 74.02±1.90 | 75.69±2.00 | 87.16±0.37 | 80.81±4.75 | 85.29±3.31 | 81.89±6.40 | 36.53±0.70 |
| GCN | 76.50±1.36 | 86.98±1.27 | 88.42±0.50 | 55.14±5.16 | 51.76±3.06 | 60.54±5.30 | 27.32±1.10 |
| GAT | 76.55±1.23 | 87.30±1.10 | 86.33±0.48 | 52.16±6.63 | 49.41±4.09 | 61.89±5.05 | 27.44±0.89 |
| GCNII | 77.33±1.48 | 88.37±1.25 | 90.15±0.43 | 77.57±3.83 | 80.39±3.40 | 77.86±3.79 | 37.44±1.30 |
| GCNet | 74.29±0.50 | 86.13±0.38 | 86.29±0.09 | 72.54±1.66 | 66.75±2.81 | 73.56±3.95 | 27.66±0.20 |
| FAGCN | 74.01±1.85 | 86.34±0.67 | 76.57±1.88 | 77.56±6.11 | 79.41±6.55 | 78.64±5.47 | 34.85±1.61 |
| H$^2$GCN | 77.11±1.57 | 87.87±1.20 | 89.49±0.38 | 84.86±7.23 | 87.65±4.98 | 82.70±5.28 | 35.70±1.00 |
| GPR-GNN | 77.13±1.67 | 87.95±1.18 | 87.54±0.38 | 78.38±4.36 | 82.94±4.21 | 80.27±8.11 | 34.63±1.22 |
| GGCN | 77.14±1.45 | 87.95±1.05 | 89.15±0.37 | 84.86±4.55 | 86.86±3.29 | 85.68±6.63 | 37.54±1.56 |
| ACM-GCN | 77.32±1.70 | 87.91±0.95 | 90.00±0.52 | 87.84±4.40 | 88.43±3.22 | 85.14±6.07 | 36.28±1.09 |
| LRGNN | 77.53±1.31 | 88.33±0.89 | **90.24±0.64** | 90.27±4.49 | 88.23±3.54 | 86.48±5.65 | 37.34±1.78 |
| PCNet | 77.50±1.06 | 88.41±0.66 | 89.51±0.28 | 88.11±2.17 | 88.63±2.75 | 82.61±2.70 | **37.80±0.64** |
| **DSR-GNN** | **78.38±0.81** | **88.64±0.61** | 89.58±0.15 | **92.61±2.98** | **90.60±1.80** | **90.50±2.79** | 37.57±0.81 |

- The highest and second-highest results are achieved by models specifically designed to handle heterophilous graphs, suggesting that effectively harnessing heterophilous links can significantly improve model performance. Notably, DSR-GNN surpasses these models by successfully balancing propagation capabilities with the inherent heterophily of the graph.

## 5.3 (Q2 & Q3) Ablation Study

To demonstrate the impact of signed edges on model performance, we evaluate the role of different adjacency matrices. We first construct JGNN, a variant of DSR-GNN that omits structural refinement. In Figure 1, "with Ori. Adj." denotes JGNN utilizing the original adjacency matrix without signed edges. "with Cos." and "with Pre." indicate JGNN using signs generated by the cosine similarity and the proposed pre-training method, respectively. The figure highlights several key points: 1) We note that JGNN with Pre. achieves superior performance across most datasets, particularly on heterophilous graphs. 2) Assigning signs to edges helps the model to distinguish neighbors, significantly enhancing the model's discriminative power. 3) JGNN with Pre. substantially outperforms JGNN using cosine similarity on both homophilous and heterophilous

graphs. The suboptimal performance of JGNN with cosine similarity can be attributed to inaccuracies in sign prediction caused by noise in the raw data. Moreover, we observe that predicting edge signs using similarity shows only marginal improvements over the original adjacency matrix, and in some cases, even leads to performance degradation. This indicates that incorrectly assigning a negative/positive sign to a homophilous/heterophilous edge can adversely affect model performance.

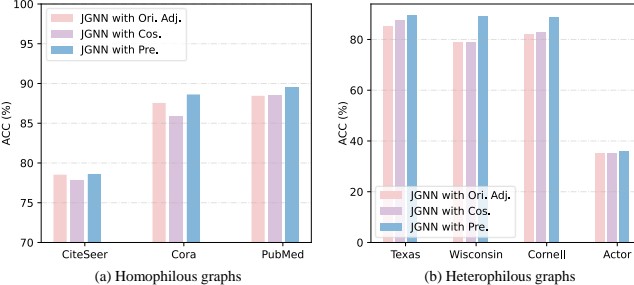

(a) Homophilous graphs    (b) Heterophilous graphs

**Figure 1: Performance of JGNN using various adjacency matrices on homophilous and heterophilous datasets, where JGNN is DSR-GNN w/o structural refinement.**

**Table 3: Ablation study: Mean accuracy (%) ± Standard deviation (%). Best performance is highlighted in bold.**

| Datasets | Structural refinement | Aggregated-feature distance | Homophily difference | ACC | Datasets | Structural refinement | Aggregated-feature distance | Homophily difference | ACC |
|---|---|---|---|---|---|---|---|---|---|
| Citeseer | | | | **78.62±0.54** | Cora | | | | 88.62±0.83 |
| | ✓ | | | 77.44±0.88 | | ✓ | | | 87.31±0.59 |
| | ✓ | ✓ | | 78.30±0.88 | | ✓ | ✓ | | 88.31±0.49 |
| | ✓ | ✓ | ✓ | 78.38±0.81 | | ✓ | ✓ | ✓ | **88.64±0.61** |
| Texas | | | | 89.50±3.30 | Wisconsin | | | | 89.20±1.60 |
| | ✓ | | | 90.75±3.30 | | ✓ | | | 88.20±1.40 |
| | ✓ | ✓ | | 92.89±3.56 | | ✓ | ✓ | | 89.60±1.50 |
| | ✓ | ✓ | ✓ | **92.61±2.98** | | ✓ | ✓ | ✓ | **90.60±1.80** |
| Cornell | | | | 88.94±3.61 | Actor | | | | 36.06±0.83 |
| | ✓ | | | 90.78±3.77 | | ✓ | | | 36.01±0.94 |
| | ✓ | ✓ | | 90.00±2.79 | | ✓ | ✓ | | 36.92 (0.79) |
| | ✓ | ✓ | ✓ | **90.50±2.79** | | ✓ | ✓ | ✓ | **37.57 (0.81)** |

To validate the role of disparity-induced structural refinement, we conduct an ablation study of each focal component, as displayed in Table 3. When DSR-GNN solely uses the signed adjacency matrix obtained by the pre-training procedure for message passing, performance declines, particularly on heterophilous graphs. However, this variant still outperforms other competitors on some datasets (e.g., Texas) due to the effective pre-training scheme. Subsequently, we incorporate structural refinement through random edge dropping, which positively impacts the model but still leaves room for further enhancement. Observations reveal that eliminating some heterophilous links to rationally balance graph heterophily with graph topology can optimize the embedding generated by DSR-GNN. Moreover, the disparity score with only the aggregated-feature distance provides minimal benefits, as it considers feature-level relationships but neglects structural information. In brief, superior accuracy is obtained by the model combining three components. It is worth noting in the table that on homophilous datasets Cora and Citeseer, due to clear connections between nodes of the same class, signed edges assisting nodes to distinguish inter-class information have allowed the model to achieve comparable performance.

## 5.4 (Q4) Comparison of Original and Refined Graphs

To highlight the differences between the refined and original graphs, we compare heterophily ratios of various datasets across distinct layers, as shown in Table 4. The data reveal that the heterophily ratio fluctuates on most datasets rather than showing a consistent decline, which indicates that DSR-GNN accomplishes refinement based on high-order disparity score instead of merely removing heterophilous edges. Meanwhile, as shown in Table 3, the performance achieved by the original graph is suboptimal compared to that of the refined graph, which may remove homophilous edges. These phenomenons illustrate that the influence of homo./hete. links on performance is not strictly positive/negative. The proposed structural refinement scheme effectively balances both types of links, thereby mitigating the adverse effects of extraneous high-order information.

Moreover, to intuitively compare the original and refined graphs, Figure 2 visualizes the graph structures used by DSR-GNN in the

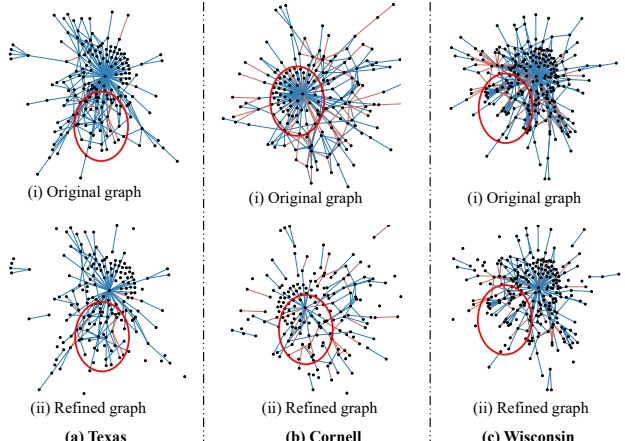

**Figure 2: Visualizations of the original graph and the refined graph used in 4th layer on the (a) Texas, (b) Cornell and (c) Wisconsin datasets, respectively. Here, blue/red lines indicate heterophilous/homophilous links, and the red circles highlight areas where significant changes occur between them.**

4th layers on the Texas, Cornell and Wisconsin datasets, respectively. Initially, it is evident that heterophilous links outnumber homophilous links in these datasets. However, due to their pronounced heterophily, the number of heterophilous links is notably reduced after the refinement process.

## 5.5 (Q5) Visualization of Layer-wise Weights $\{\lambda_l\}_{l=0}^{L}$

To intuitively understand the impact of high-order information on the gained representations, we visualize the learned weights $\{\lambda_l\}_{l=0}^{4}$ of DSR-GNN with four convolutional layers on several datasets, as illustrated in Figure 3. From this figure, we observe that for three heterophilous graphs (Texas, Cornell and Winconsin), the weights assigned to neighbors decrease as the number of hops increases. Although the structural refinement operation allows the model to aggregate high-order neighbors with minimal disparity scores, they

**Table 4: Heterophily ratio of various graphs varies across different layers, where the heterophily ratio in the $l$-th refined graph $\mathcal{G}^{(l)}$ is $\mathcal{H}^{-}(\mathcal{G}^{(l)}) = 1 - \frac{\sum_{i=1}^{N} h_i^{(l)}}{N}$.**

| Datasets/Hete. ratio | Citeseer | Cora | Pubmed | Texas | Wisconsin | Cornell | Actor |
|---|---|---|---|---|---|---|---|
| $\mathcal{H}^{-}(\mathcal{G})$ | 0.2609 | 0.1900 | 0.1976 | 0.8923 | 0.8039 | 0.6946 | 0.7812 |
| $\mathcal{H}^{-}(\mathcal{G}^{(2)})$ | 0.2883 | 0.1648 | 0.2139 | 0.8684 | 0.8497 | 0.6810 | 0.7612 |
| $\mathcal{H}^{-}(\mathcal{G}^{(3)})$ | 0.2908 | 0.1746 | 0.2143 | 0.8776 | 0.8507 | 0.6781 | 0.7606 |
| $\mathcal{H}^{-}(\mathcal{G}^{(4)})$ | 0.2906 | 0.1758 | 0.2140 | 0.8585 | 0.8231 | 0.6735 | 0.7624 |

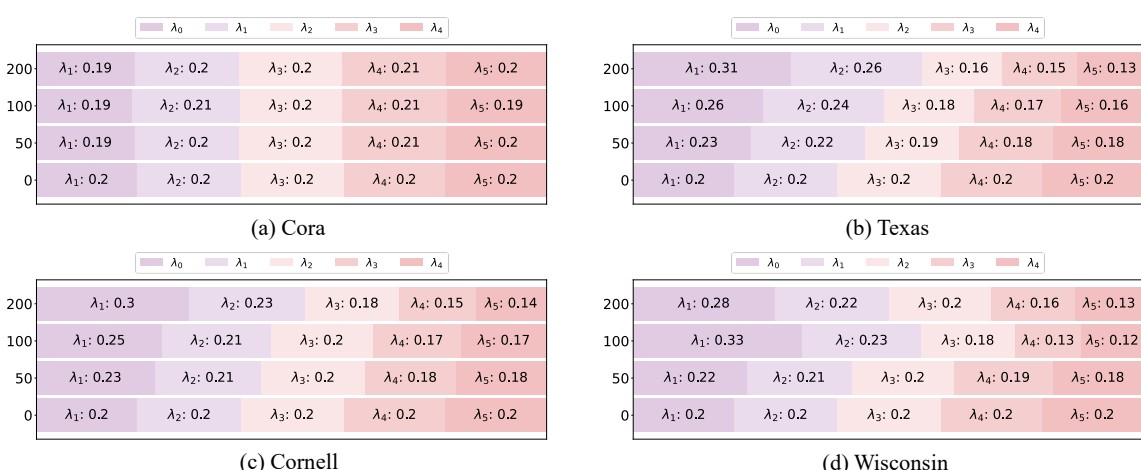

(a) Cora  (b) Texas

(c) Cornell  (d) Wisconsin

**Figure 3: Visualizations of learnable layer-wise weights $\{\lambda_l\}_{l=0}^{4}$ of 4-layer DSR-GNN, where the vertical axis represents the number of epochs.**

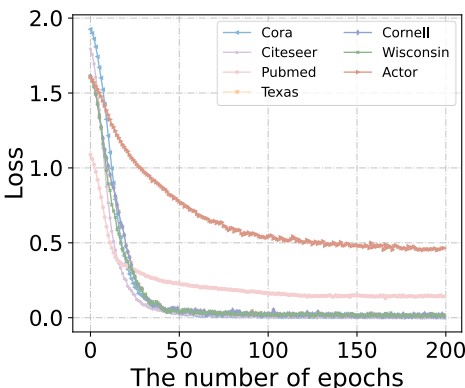

**Figure 4: Loss curves during the training procedure of DSR-GNN on seven datasets.**

provide less information and thus receive lower weight. Notably, the significance of 3-hop and 4-hop interactions is quite similar, suggesting that the model effectively mitigates the incorporation of noise as the number of hops increases. For the homophilous graph Cora, the weights assigned to each hop are more balanced, indicating a strong feature similarity between nodes. This characteristic facilitates consistent message passing across layers.

Moreover, the loss values of DSR-GNN across seven datasets during the training process are depicted in Figure 4. Notably, the losses decrease significantly throughout the training epochs, followed by a gradual stabilization. This phenomenon underscores that the model is effective and can achieve stable state through continuous optimization.

## 6 CONCLUSION

In this paper, we proposed a novel framework that integrated a Disparity-induced Structural Refinement (DSR) scheme with Graph Neural Network (GNN), termed DSR-GNN, to enhance representation learning on heterophilous graphs. The model incorporated two collaborative steps to optimize message propagation and fusion. In specific, the initial step designed a disparity score, derived from the theory of error bound for node classification, to evaluate high-order relationships between nodes based on both features and structure information. The score derived the construction of layer-wise edges by eliminating links with significant disparity, thereby minimizing the impact of irrelevant high-order information during message passing. Meanwhile, the gained node representations can optimize the disparity score in return. Extensive experiments of the proposed model on both heterophilous and homophilous datasets demonstrated that DSR-GNN outperformed existing methods, showcasing its effectiveness in handling heterophilous links.

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

# A APPENDIX

## A.1 Proofs

DEFINITION 5. *Let's define a L-layer GNN classifier $f$, for node $v_i$, the prediction score is $f_i(\mathbf{X}, \mathcal{G}) = f(g_i(\mathbf{X}, \mathcal{G}); \Theta^{(1)}, \Theta^{(2)}, \cdots, \Theta^{(L)})$, where $g$ denotes a feature aggregation function and $f$ is a ReLU-activated L-layer MLP with learnable parameters $\{\Theta^{(l)}\}_{l=1}^L$. We assume that the maximum number of hidden units across all layers is $b$.*

DEFINITION 6. *For any node $v_i$, the distance of aggregated features from it to other node $v_j$ is defined as*

$$\epsilon_{ij} = \|g_i(\mathbf{X}, \mathcal{G}) - g_j(\mathbf{X}, \mathcal{G})\|_2. \tag{12}$$

DEFINITION 7. *Given a labeled node $v_j \in V_{lab}$ with label $y_j$, there exists a margin $\gamma \geq 0$ satisfing*

$$f_j(\mathbf{X}, \mathcal{G})[y_j] \leq \gamma + max_{c \neq y_j} f_j(\mathbf{X}, \mathcal{G})[c], \tag{13}$$

*where $f_j(\mathbf{X}, \mathcal{G})[\cdot]$ is to take an element of the predicted probability vector (w.r.t classifier).*

DEFINITION 8. *The expected loss $\mathcal{L}_i^\gamma(f)$ of the classifier $f$ on $v_i$ for a margin $\gamma$ and any distribution $\mathcal{D}$ is defined as [25, 28]:*

$$\mathcal{L}_i^\gamma(f) := \mathbb{P}_{v_i \sim \mathcal{D}}\left[f_i(\mathbf{X}, \mathcal{G})[y_i] \leq \gamma + max_{c \neq y_i} f_i(\mathbf{X}, \mathcal{G})[c]\right]. \tag{14}$$

*The empirical loss is denoted as $\hat{\mathcal{L}}_i^\gamma(f)$ that is the empirical estimate of the expected loss.*

ASSUMPTION 1. *Let $P$ be a distribution on the classifier family $\mathcal{F}$, defined by sampling the vectorized MLP parameters from $\mathcal{N}\left(0, \sigma^2 I\right)$ for some $\sigma^2 \leq \frac{(\gamma/8\epsilon_{ij})^{2/L}}{2b(\lambda + \ln 2bL)}$.*

LEMMA 1. *(Lemma 2 in [26]) With assumptions (1) A balance class distribution with $\mathbf{P}(\mathbf{Y} = 1) = \mathbf{P}(\mathbf{Y} = 0)$ and (2) Aggregated feature distribution shares the same variance $\sigma\mathbf{I}$. When nodes $v_i$ and $v_j$ have the same aggregated features $\|\mathbf{f}_i - \mathbf{f}_j\| = \epsilon_{ij}$, we can have:*

$$\left|\mathbf{P}_1\left(y_i = c_1 \mid \mathbf{f}_i\right) - \mathbf{P}_2\left(y_j = c_1 \mid \mathbf{f}_j\right)\right| \leq$$
$$\frac{\rho}{\sqrt{2\pi}\sigma}\left(\epsilon_{ij} + \rho\left|h_i^+ - h_j^+\right|\right), \tag{15}$$

*where $\rho = \|\boldsymbol{\mu}_1 - \boldsymbol{\mu}_2\|$ is original feature separability of nodes, $\mathbf{P}_1$ and $\mathbf{P}_2$ are the conditional probability and $h_i^+$ denotes the node-level homophily ratio of node $v_i$. Specifically, the node features follow the Gaussian distribution: $\mathbf{x}_i \sim N(\boldsymbol{\mu}_1, \mathbf{I})$ for $i \in V_{lab}$ and $\mathbf{x}_i \sim N(\boldsymbol{\mu}_2, \mathbf{I})$ for $i \notin V_{lab}$.*

THEOREM 2. *(Node Pair Generalization of Deterministic Classifiers [25]). Let $\widetilde{f}$ be any classifier in $\mathcal{F}$. For any node $v_i$, for any $\lambda > 0$ and $\gamma \geq 0$, for any "prior" distribution $P$ on $\mathcal{F}$ that is independent of the training data $v_j$, with probability at least $1 - \delta$ over the sample of $y_j$, for any $Q$ on $\mathcal{F}$ such that $\mathrm{Pr}_{f \sim Q}\left(\left\|f_i(X, G) - \widetilde{f}_i(X, G)\right\|_\infty < \frac{\gamma}{8}\right) > \frac{1}{2}$, we have*

$$\mathcal{L}_i^0(\widetilde{f}) \leq \widehat{\mathcal{L}}_j^\gamma(\widetilde{f}) + \frac{1}{\lambda}\left(2\left(D_{\mathrm{KL}}(Q\|P) + 1\right) + \ln\frac{1}{\delta} + \frac{\lambda^2}{4}\right. \tag{16}$$
$$\left. + \ln\mathbb{E}_{f \sim P}e^{\lambda(\mathcal{L}_i^{\gamma/4}(f) - \mathcal{L}_j^{\gamma/2}(f))}\right)$$

LEMMA 2. *Suppose an L-layer GNN classifier $f$ is associated with model parameters $\Theta^{(1)}, \ldots, \Theta^{(L)}$. Define $T_f := \max_{l=1,\ldots,L}\left\|\Theta^{(l)}\right\|_2$. For any node $v_i$ and $\gamma \geq 0$, if $\epsilon_{ij}T_f^L \leq \frac{\gamma}{4}$, then*

$$\mathcal{L}_i^{\gamma/2}(f) - \mathcal{L}_j^\gamma(f) \leq \frac{C\rho}{\sqrt{2\pi}\sigma}(\epsilon_{ij} + |h_i^+ - h_j^+|\rho). \tag{17}$$

PROOF. We denote $f_i$ as $f_i(\mathbf{X}, \mathcal{G})$ and $\eta_c(i)$ as $Pr(y_i = c|g_i(\mathbf{X}, \mathcal{G}))$. Following the above analysis, we have

$$\mathcal{L}_i^{\gamma/2}(f) - \mathcal{L}_j^\gamma(f)$$
$$=\mathbb{E}_{y_i}\mathcal{L}^{\gamma/2}(f_i, y_i) - \mathbb{E}_{y_j}\mathcal{L}^\gamma(f_j, y_j)$$
$$=\sum_{c=1}^C \eta_c(i)\mathcal{L}^{\gamma/2}(f_i, c) - \sum_{c=1}^C Pr(y_j = c)\mathcal{L}^\gamma(f_j, c)$$
$$=\sum_{c=1}^C \left(\eta_c(i)\mathcal{L}^{\gamma/2}(f_i, c) - \eta_c(j)\mathcal{L}^\gamma(f_j, c)\right)$$
$$=\sum_{c=1}^C \left(\eta_c(i)(\mathcal{L}^{\gamma/2}(f_i, c) - \mathcal{L}^\gamma(f_j, c))\right.$$ 
$$\left. + (\eta_c(i) - \eta_c(j))\mathcal{L}^\gamma(f_j, c)\right)$$
$$\leq\sum_{c=1}^C \left((\mathcal{L}^{\gamma/2}(f_i, c) - \mathcal{L}^\gamma(f_j, c)) + (\eta_c(i) - \eta_c(j))\right). \tag{18}$$

According to Lemma 1, we have

$$\eta_c(i) - \eta_c(j) \leq \frac{\rho}{\sqrt{2\pi}\sigma}(\epsilon_{ij} + |h_i^+ - h_j^+|\rho). \tag{19}$$

Moreover, we have

$$\|f_i - f_j\|_\infty \leq \frac{\gamma}{4}. \tag{20}$$

Therefore, we can rewrite it as follows

$$f_i(\mathbf{X}, \mathcal{G})[y_i] - f_j(\mathbf{X}, \mathcal{G})[y_j] \leq \frac{\gamma}{4}. \tag{21}$$

According to the definition of Expected Margin Loss, we have

$$\mathcal{L}^{\gamma/2}(f_i, c) \leq \mathcal{L}^\gamma(f_j, c), \tag{22}$$

Consequently, the original bound can be scaled as

$$\mathcal{L}_i^{\gamma/2}(f) - \mathcal{L}_j^\gamma(f) \leq \frac{C\rho}{\sqrt{2\pi}\sigma}(\epsilon_{ij} + |h_i^+ - h_j^+|\rho), \tag{23}$$

which completing the proof. □

LEMMA 3. *For any node $v_i$, any $\lambda > 0$ and $\gamma \geq 0$, assume the "prior" $P$ on $\mathcal{F}$ is defined by sampling the vectorized parameters from $\mathcal{N}\left(0, \sigma^2 I\right)$ for some $\sigma^2 \leq \frac{(\gamma/8\epsilon_{ij})^{2/L}}{2b(\lambda + \ln 2bL)}$. We have*

$$\ln\mathbb{E}_{f \sim P}e^{\lambda(\mathcal{L}_i^{\gamma/4}(f) - \mathcal{L}_j^{\gamma/2}(f))} \leq \ln 3 + \frac{C\rho}{\sqrt{2\pi}\sigma}(\epsilon_{ij} + |h_i^+ - h_j^+|\rho). \tag{24}$$

PROOF. Under the condition in Lemma 2, we can split the classifier's space into two regimes. (a): $Pr(\epsilon_{ij}T_f^L \leq \frac{\gamma}{8})$ and (b): $Pr(\epsilon_{ij}T_f^L > \frac{\gamma}{8})$.

Firstly, by Lemma 2, we have $e^{\lambda(\mathcal{L}_i^{\gamma/4}(f) - \mathcal{L}_j^{\gamma/2}(f))} \leq e^{\frac{\lambda C\rho}{\sqrt{2\pi}\sigma}(\epsilon_{ij} + |h_i^+ - h_j^+|\rho)}$ for any $\epsilon_{ij}T_f^L \leq \frac{\gamma}{8}$. Then, for $\epsilon_{ij}T_f^L > \frac{\gamma}{8}$, according to Assumption 3 in [25], with probability at least $1 - e$,

$$e^{\lambda\left(\mathcal{L}_i^{\gamma/4}(f) - \mathcal{L}_j^{\gamma/2}(f)\right)} \leq e^{\lambda + \frac{\lambda C\rho}{\sqrt{2\pi}\sigma}(\epsilon_{ij} + |h_i^+ - h_j^+|\rho)}. \tag{25}$$

Notably, this assumption is satisfied since we are considering relationships between pairs of nodes in this paper. Moreover, according to [33] and under the condition $\sigma^2 \leq \frac{(\gamma/8\epsilon_{ij})^{2/L}}{2b(\lambda+\ln 2bL)}$, we have the following inequality:

$$Pr(\epsilon_{ij}T_f^L > \frac{\gamma}{8}) \leq e^{-\lambda}. \tag{26}$$

Therefore, we have

$$\ln \mathbb{E}_{f \sim P} e^{\lambda(\mathcal{L}_i^{\gamma/4}(f) - \mathcal{L}_j^{\gamma/2}(f))}$$

$$\leq \ln \left( Pr(\epsilon_{ij}T_f^L > \frac{\gamma}{8})\left((1-e^{-1})e^{\lambda+\frac{\lambda C\rho}{\sqrt{2\pi}\sigma}(\epsilon_{ij}+|h_i^+-h_j^+|\rho)}\right.\right.$$

$$\left.\left. + e^{-1}e^\lambda \right) + Pr(\epsilon_{ij}T_f^L \leq \frac{\gamma}{8})e^{\frac{\lambda C\rho}{\sqrt{2\pi}\sigma}(\epsilon_{ij}+|h_i^+-h_j^+|\rho)} \right)$$

$$\leq \ln \left( Pr(\epsilon_{ij}T_f^L > \frac{\gamma}{8})e^{\lambda+\frac{\lambda C\rho}{\sqrt{2\pi}\sigma}(\epsilon_{ij}+|h_i^+-h_j^+|\rho)} + e^{\lambda-1} \right.$$

$$\left. + e^{\frac{\lambda C\rho}{\sqrt{2\pi}\sigma}(\epsilon_{ij}+|h_i^+-h_j^+|\rho)} \right) \tag{27}$$

$$\leq \ln \left( e^{-\lambda}e^{\lambda+\frac{\lambda C\rho}{\sqrt{2\pi}\sigma}(\epsilon_{ij}+|h_i^+-h_j^+|\rho)} + e^{\lambda-1} \right.$$

$$\left. + e^{\frac{\lambda C\rho}{\sqrt{2\pi}\sigma}(\epsilon_{ij}+|h_i^+-h_j^+|\rho)} \right)$$

$$\leq \ln \left( 1 + e^{\frac{2C\rho}{\sqrt{2\pi}\sigma}(\epsilon_{ij}+|h_i^+-h_j^+|\rho)} \right)$$

$$\leq \ln e^{\frac{3C\rho}{\sqrt{2\pi}\sigma}(\epsilon_{ij}+|h_i^+-h_j^+|\rho)} = \ln 3 + \frac{C\rho}{\sqrt{2\pi}\sigma}(\epsilon_{ij}+|h_i^+-h_j^+|\rho).$$

In the later steps of formula derivation, we set $\lambda = 1$ since we are considering the relationships between pairs of nodes. $\square$

**Theorem 3 (Error Bound for Node Classification).** *Let $f$ be a classifier in the classifier family $\mathcal{F}$ with learnable parameters $\{\Theta^{(l)}\}_{l=1}^L$ that conform with the normal distribution, then for any node $v_i$ and $\gamma \geq 0$, we have*

$$\mathcal{L}_i^0(f) \leq \widehat{\mathcal{L}}_j^\gamma(f) + O\left( \frac{C\rho}{\sqrt{2\pi}\sigma}(\epsilon_{ij}+\rho|h_i^+-h_j^+|) \right.$$

$$\left. + \frac{\sum_{l=1}^L \|\Theta^{(l)}\|_F^2}{\sigma^2} \right), \tag{28}$$

*where* $\sigma = \min \left( \frac{(\gamma/8\epsilon_{ij})^{1/L}}{\sqrt{2b(1+\ln(2bL))}}, \frac{\gamma}{84LB_i\beta^{L-1}\sqrt{b\ln(4bL)}} \right)$, $B_i = \|g_i(\mathbf{X}, \mathcal{G})\|_2$, $h_i^+$ *denotes the homophily ratio of node $v_i$ and $\rho$ is original feature separability of nodes.*

**Proof.** According to Theorem 1 and Lemma 3, we have the following inequality:

$$\mathcal{L}_i^0(f) \leq \widehat{\mathcal{L}}_j^\gamma(f) + \frac{1}{\lambda}\left( 2\left(D_{\mathrm{KL}}(Q\|P) + 1\right) + \ln\frac{1}{\delta} + \frac{\lambda^2}{4} \right.$$

$$\left. + \ln 3 + \frac{C\rho}{\sqrt{2\pi}\sigma}(\epsilon_{ij}+|h_i^+-h_j^+|\rho) \right) \tag{29}$$

As before, we set $\lambda = 1$ since we are considering the relationships between pairs of nodes. Therefore, we can rewritten this inequality as follows:

$$\mathcal{L}_i^0(f) \leq \widehat{\mathcal{L}}_j^\gamma(f) + \left( 2(D_{\mathrm{KL}}(Q\|P) + 1) + \frac{1}{4} + \ln\frac{3}{\delta} \right.$$

$$\left. + \frac{C\rho}{\sqrt{2\pi}\sigma}(\epsilon_{ij}+|h_i^+-h_j^+|\rho) \right) \tag{30}$$

Moreover, according to [33], $P$ and $Q$ are normal distributions, we have

$$D_{KL}(Q\|P) \leq \frac{\sum_{l=1}^L \|\Theta^{(l)}\|_F^2}{\sigma^2}, \tag{31}$$

where $\sigma = \min \left( \frac{(\gamma/8\epsilon_{ij})^{1/L}}{\sqrt{2b(1+\ln(2bL))}}, \frac{\gamma}{84LB_i\beta^{L-1}\sqrt{b\ln(4bL)}} \right)$. Consequently, we derive the following bound corresponding to pairs of nodes:

$$\mathcal{L}_i^0(f) \leq \widehat{\mathcal{L}}_j^\gamma(f) + O\left( \frac{C\rho}{\sqrt{2\pi}\sigma}(\epsilon_{ij}+\rho|h_i^+-h_j^+|) \right.$$

$$\left. + \frac{\sum_{l=1}^L \|\Theta^{(l)}\|_F^2}{\sigma^2} \right). \tag{32}$$

Note that the above derivation process can be applied to the case of unlabeled nodes (*i.e.* for any unlabeled node $v_i$). $\square$

