# OpenReview forum: "Boosting Graph Convolution with Disparity-induced Structural Refinement"
_ACM.org/TheWebConf/2025/Conference — WWW 2025 Poster_

### Official Review · Reviewer_vvDd · 2024-11-29

**Novelty:** 5
**Technical Quality:** 6

**Review:**

The paper presents a novel framework, DSR-GNN, designed to enhance graph neural networks (GNNs) in scenarios involving both homophilous and heterophilous graphs. The technical contribution lies in the Disparity-Induced Structural Refinement (DSR), which adaptively refines graph structures during message passing by leveraging a disparity score. This score integrates feature similarity and structural characteristics to mitigate the influence of irrelevant or misleading information. Theoretical analysis supports the method, and empirical evaluations demonstrate superior performance compared to state-of-the-art approaches on benchmark datasets.

**Strengths:**
- Techniques: The proposed disparity-induced structural refinement addresses a critical limitation of existing GNNs in handling heterophilous graphs, with a solid theoretical underpinning.

- Theoretical guarantee: The authors derive error bounds for node classification, grounding the design decisions in theoretical insights. This level of rigor distinguishes the paper from empirical-only approaches.

- Experiments: The evaluation spans diverse datasets with varying levels of homophily, providing strong evidence of the method's robustness and generality. The ablation studies are detailed, isolating the impact of each component.

- Writing: The paper is well-structured, with clear problem statements, explanations of methodology, and discussions of results.

**Weaknesses:**

- Literature review. The authors still miss some important related works in this field of Homo/Heterophile graph learning, e.g., [1] for classification, and [2] for dynamic graph learning. Especially [2] proposes topology remedy and layer-wise propagation with target homophily.  As they are both focusing on the topology optimization towards targets,  differences should be distinguished.

- Significance tests. Even the proposed DSR shows improvements (achieve best results 5/7), the significance tests should be applied for further evaluation.

- Interpretability of  Layer-wise Weight Learning: The importance of layer-wise weights $\lambda$ is demonstrated, but their interpretability and convergence properties are not extensively analyzed.

- Efficiency issue: While the paper mentions GPU acceleration, there is no analysis of the computational overhead introduced by the disparity score computation and graph refinement, particularly extending to large-scale graphs. More datasets on larger graphs should be incorporated.

[1] Bo D, Wang X, Shi C, et al. Beyond low-frequency information in graph convolutional networks[C]//Proceedings of the AAAI conference on artificial intelligence. 2021, 35(5): 3950-3957.
[2] Zhou Z, Huang Q, Lin G, et al. Greto: Remedying dynamic graph topology-task discordance via target homophily[C]//The eleventh international conference on learning representations. 2023.

**Questions:**

Please refer to my weaknesses.

- Analyze more related literature and distinguish their differences.

- Provide significant tests on results can better show the improvement.

- Efficiency analysis should be provided and more discussions on how to adapt to large-scale graphs should be provided.

- Provide some theoretical interpretability of  Layer-wise Weight Learning and show differences against  GReTo[2].

**Reviewer Confidence:**

4: The reviewer is certain that the evaluation is correct and very familiar with the relevant literature

**Scope:**

3: The work is somewhat relevant to the Web and to the track, and is of narrow interest to a sub-community

---

### Official Review · Reviewer_5Bgw · 2024-12-01

**Novelty:** 5
**Technical Quality:** 4

**Review:**

Advantages:

1. The paper proposes the innovative Disparity-induced Structural Refinement (DSR) method to address the performance limitations of Graph Neural Networks (GNNs) on heterogeneous graphs, effectively mitigating the information interference caused by heterogeneous connections.

2. The experimental section comprehensively covers multiple real-world datasets and compares the proposed method with various existing approaches, demonstrating the superior performance of DSR-GNN.

3. DSR combines node features and structural information, intelligently adjusting the message-passing process, with both theoretical and experimental results complementing each other.

Disadvantages:

1. Although the paper introduces the Disparity Score, the theoretical discussion lacks depth. For example, there is insufficient comparison with other existing graph structure optimization methods in terms of their fundamental differences.
2. The experiments do not provide a detailed discussion of the impact of hyperparameters (e.g., the number of layers, the number of neighboring nodes) on model performance, which may affect the reproducibility and applicability of the method.
3. The literature review on similar heterogeneous graph learning methods is not comprehensive, with some important works (e.g., recent GNN improvement methods) not being mentioned.

Comments:

1. It is suggested to further elaborate on the mathematical background of the Disparity Score or its key contribution to enhancing the model's representational ability. For instance, can the impact of different Disparity Score thresholds on model performance be quantified?
2. Although the paper conducts experiments on multiple datasets, it lacks an ablation study of the individual components of DSR, making it difficult to clearly identify the contribution of each module.
3. The comparison with existing methods mainly focuses on experimental performance. It is recommended to also discuss the design philosophy and the scope of application of the method.
4. Some of the graphs (e.g., Table 2, Figure 1) have small font sizes, which hinder readability. It is suggested to adjust the layout to improve readability.
5. The paper primarily focuses on academic scenarios. It would be helpful to discuss the potential application value of DSR-GNN in real-world industrial scenarios in the conclusion.

**Questions:**

1. Is the design of DSR-GNN suitable for dynamic or multimodal graphs? If not, what are the limitations or potential improvements?

2. Has the choice of Disparity Score thresholds been systematically validated in terms of its impact on model performance? Are there any recommended default values?

3. How efficient is DSR-GNN in terms of training on heterogeneous graphs? Does it incur additional computational overhead compared to other mainstream methods?

**Reviewer Confidence:**

2: The reviewer is willing to defend the evaluation, but it is likely that the reviewer did not understand parts of the paper

**Scope:**

3: The work is somewhat relevant to the Web and to the track, and is of narrow interest to a sub-community

---

### Official Review · Reviewer_izth · 2024-12-02

**Novelty:** 4
**Technical Quality:** 5

**Review:**

The authors addressed the limitations of previous heterophilous graph methods, which relied solely on node features to infer node relationships while neglecting structural information and suffered from estimation inaccuracies due to ratio of heterophily. They proposed a disparity-induced structural refinement (DSR) method to enhance the representation capability of heterophilous graphs. By effectively identifying and leveraging the potential advantages of heterophilous edges, the authors designed a disparity score based on the derivation of node classification error bounds. This score integrates feature and structural information to represent aggregation distance and heterophily rate differences, regulating neighbor aggregation while mitigating irrelevant information during message passing. Experiments demonstrate the effectiveness of the DSR-GNN method on heterophilous datasets, validating the utility of signed edges, structural refinement, and disparity scores.

Strengths

1. The authors clearly articulated the problem, making the technical approach easy to understand. The theoretical analysis is comprehensive, and the writing is coherent.

2. The paper provides detailed proof to facilitate a better understanding of the mechanism for the readers.

Weaknesses

Although the paper offers detailed proofs, some theorems could benefit from more intuitive explanations to enhance their clarity and reasoning.

**Questions:**

1. Disparity scores are used to guide the removal of “irrelevant” higher-order information while retaining higher-order neighbors with minimal disparity. Is it possible that the removed “irrelevant” higher-order information could also contain useful data?

2. Could the authors discuss the model’s scalability and computational complexity, such as its performance on the ogbn-arxiv and arXiv-year datasets?

3. The provided link to the code appears to be inaccessible at the moment. It might be helpful to verify the link or provide an alternative way to access the code to ensure reproducibility.

**Reviewer Confidence:**

4: The reviewer is certain that the evaluation is correct and very familiar with the relevant literature

**Scope:**

4: The work is relevant to the Web and to the track, and is of broad interest to the community

---

### Official Review · Reviewer_cjt8 · 2024-12-03

**Novelty:** 4
**Technical Quality:** 4

**Review:**

This paper presents a model of graph convolution enhanced with disparity-induced structural refinement. It assumes that the classification of nodes can not only be influenced by the distance of aggregated feature between nodes, but also the disparity in homophily ratios. The paper focuses on incorporating the latter factor into the model.
The paper proves that the classification loss is indeed influenced by the distance of aggregated feature and the disparity in homophily ratios. This provides a basis for the development of the algorithm. Specific assignment of edges and message propagation are proposed in the network.
The model is tested on several Homo/Heterophile datasets. It is compared to several existing baselines. The general observation is that the proposed method can slightly improve the accuracy over the baselines (but not always).
The idea to take into account the homophily ratios and use them to adjust message propagation in the network is interesting. Although similar idea has been used in some of the baseline methods, the novelty in this paper lies in its specific way to estimate homophily ratios and to design a message propagation method accordingly.
The experiments are extensive. However, the advantage of the proposed method is not always demonstrated with respect to the baselines. On some of the datasets, the proposed method underperforms some of the baselines.
In the ablation experiment (Table 3), it is shown that the proposed structural refinement is not always helpful. This observation may contradict the initial assumption and idea. This should be further analyzed.

**Questions:**

As the behavior of the proposed method varies on different datasets, what is the characteristic of the dataset on which the proposed method is the most appropriate?

The proposed method only consider whether an edge connects two nodes of the same class or different classes. These two types of edge may have different impact on the classification of nodes, but this is strongly dependent on the application. The datasets used in the experiments are relatively simple networks in which only one relation (citation or cooccurrence) exists. In this setting, it is reasonable to assume that nodes of similar classes would be connected. How general is this assumption in network-based applications? Could the proposed method be extended to other types of heterogeneous network containing different types of relations between nodes?

Typo: Line 492: Its strategy is sample[simple] and intuitive
Suggestion: The description about the datasets can be enhanced. It is not said that the nodes are of several classes. One has to check in the referred papers for this information. The test task is also not clearly described. One can guess that it is node classification.
Some of the baselines are described with respect to the proposed method, but many other are not. It would be good to provide a short summary on the differences between the baselines and the proposed method, so that one can better understand how these differences influence the accuracy.

**Reviewer Confidence:**

3: The reviewer is confident but not certain that the evaluation is correct

**Scope:**

4: The work is relevant to the Web and to the track, and is of broad interest to the community

---

### Official Review · Reviewer_8o4t · 2024-12-11

**Novelty:** 4
**Technical Quality:** 4

**Review:**

The paper proposes a novel Dynamic Structural Refinement (DSR) method for GNNs, enabling adaptive and selective message propagation to enhance representation learning on heterophilous graphs. It provides strong theoretical support through error-bound derivations for node classification, demonstrating the necessity of structural refinement during message passing. While I am not deeply familiar with this field, the method and analysis appear innovative and robust. I recommend acceptance.

**Questions:**

NA

**Reviewer Confidence:**

1: The reviewer's evaluation is an educated guess

**Scope:**

3: The work is somewhat relevant to the Web and to the track, and is of narrow interest to a sub-community